# Simultaneous Sonochemical Coloration and Antibacterial Functionalization of Leather with Selenium Nanoparticles (SeNPs)

**DOI:** 10.3390/polym14010074

**Published:** 2021-12-25

**Authors:** Tarek Abou Elmaaty, Khaled Sayed-Ahmed, Radwan Mohamed Ali, Kholoud El-Khodary, Shereen A. Abdeldayem

**Affiliations:** 1Department of Material Art, Galala University, Galala 43713, Egypt; 2Department of Textile Printing, Dyeing and Finishing, Faculty of Applied Arts, Damietta University, Damietta 34512, Egypt; khloud.el.khodary@gmail.com (K.E.-K.); shereen.abdeldayem.82@gmail.com (S.A.A.); 3Department of Agricultural Chemistry, Faculty of Agriculture, Damietta University, Damietta 34512, Egypt; drkhaled_1@du.edu.eg; 4Department of Biochemistry, Faculty of Agriculture, Al-Azhar University, Cairo 11651, Egypt; yaseenjjar@gmail.com

**Keywords:** selenium nanoparticles, leather, sonochemical, antibacterial, toxicity, footwear

## Abstract

The development of antibacterial coatings for footwear components is of great interest both from an industry and consumer point of view. In this work, the leather material was developed taking advantage of the intrinsic antibacterial activity and coloring ability of selenium nanoparticles (SeNPs). The SeNPs were synthesized and implemented into the leather surface by using ultrasonic techniques to obtain simultaneous coloring and functionalization. The formation of SeNPs in the solutions was evaluated using UV/Vis spectroscopy and the morphology of the NPs was determined by transmission electron microscopy (TEM). The treated leather material (leather/SeNPs) was characterized by scanning electron microscopy (SEM) and energy dispersive X-ray spectroscopy (EDX). The effects of SeNPs on the coloration and antibacterial properties of the leather material were evaluated. The results revealed that the NPs were mostly spherical in shape, regularly distributed, and closely anchored to the leather surface. The particle size distribution of SeNPs at concentrations of 25 mM and 50 mM was in the range of 36–77 nm and 41–149 nm, respectively. It was observed that leather/SeNPs exhibited a higher depth of shade compared to untreated ones, as well as excellent fastness properties. The results showed that leather/SeNPs can significantly enhance the antibacterial activity against model of bacteria, including Gram-positive bacteria (*Bacillus cereus*) and Gram-negative bacteria (*Pseudomonas aeruginosa*, *Salmonella typhi* and *Escherichia coli*). Moreover, the resulting leather exhibited low cytotoxicity against HFB4 cell lines. This achievement should be quite appealing to the footwear industry as a way to prevent the spread of bacterial infection promoted by humidity, poor breathability and temperature which promote the expansion of the microflora of the skin.

## 1. Introduction

Leather is a durable and flexible material made by tanning animal rawhide and skin, most commonly cattle hide [1]. It has unique characteristics, including high tensile strength, elasticity, tear resistance, high porosity, and air/water permeability [2,3]. In addition, collagen is the main component of natural leather and amino acids are the main units of collagen. It contains functional groups such as –NH_4_^+^, –COO^−^ as well as OH groups [4]. Collagen of leather can provide ideal conditions such as moisture, temperature, oxygen, and nutrients required for the rapid growth of bacteria and fungi [5].

Leathers have undergone many developments concerning their color and acquired functionalization such as antibacterial, self-cleaning, and UV protection [6,7]. In addition, diverse dyes and dyeing methods have been utilized for sustainable development in the leather industry [8]. The most important of these developments is the use of nanotechnology, which has an eco-friendly effect on the environment [9,10].

Nowadays, coloration and functionalization of leather surfaces with organic nanoparticles have received considerable attention [7,11,12,13,14,15,16]. The multifunctional features of leather, such as antimicrobial and UV radiation protection, as well as self-cleaning capabilities, have improved leather’s adaptability [1,6,9,17,18,19].

Recently, many studies have reported the use of AgNPs to functionalize leather surfaces due to their antibacterial properties [5,20,21,22,23]. In addition, silicon nanoparticles were used with polyurethanes and polyacrylates to produce leather with high water vapor permeability, good mechanical properties as well as good thermal properties [24,25]. Additionally, Ag-TiO_2_ nanoparticles were deposited on the leather surface to obtain antimicrobial properties and improve photocatalytic activity [26,27]. As well, TiO_2_-SiO_2_ nanocomposite was used to enhance the performance such as colorfastness and adhesion [28].

From what was mentioned previously, we found that the multifunctional properties of leathers improved by different kinds of nanoparticles, which were investigated to evaluate their antimicrobial, mechanical, and thermal properties, as well as UV protection and photocatalytic activity. However, the use of NPs as colorants and finishing agents in leather processing has not been sufficiently investigated. Moreover, leather finishing with NPs requires multiple steps, a long reaction time, and the use of more hazardous chemical agents. As a result, there is a great demand for a simple and environmentally safe process for the incorporation of nanoparticles into leather surfaces without the use of hazardous chemical ingredients. One of the most effective nanoparticles in this context is selenium nanoparticles (SeNPs) which have excellent coloring capability [29,30,31,32], antimicrobial, antitumor, antioxidant, and antibiofilm properties [33,34,35,36,37]. Moreover, SeNPs exhibit low cytotoxicity [29,38].

In this work we have developed a simple technique to obtain simultaneous coloration and functionalization of leather via the incorporation of SeNPs into the leather under study utilizing an ultrasonic technique. The SeNPs was synthesized through a green process and used as a colorant without any harmful additives, achieving outstanding color fastness and antibacterial properties.

To the best of our knowledge, the utilization of SeNPs for coloration and functionalization of leather has not been reported elsewhere.

## 2. Materials and Methods

### 2.1. Materials

Salted wet cowhide leathers were obtained from the Timex Tannery Company, Cairo, Egypt, and then tanned, as reported by Ali et al. [39]. As for chemicals; sodium hydrogen selenite, ascorbic acid, and polyvinylpyrrolidone (PVP) were purchased from Loba Chemie, Mumbai, India. Other chemicals were of commercial grade.

### 2.2. Green Synthesis of SeNPs

SeNPs were prepared via redox reaction as reported by Abou Elmaaty et al. [32] with an improved modification. Sodium hydrogen selenite was used at different concentrations (50 mM and 100 mM) as a precursor for SeNPs. Polyvinylpyrrolidone (PVP) was dissolved in sodium hydrogen selenite solution at a concentration of 12 g/100 mL to maintain the nanoparticles stability. Then, ascorbic acid at varied concentrations of 100 mM and 200 mM was mixed with this mixture at a volume ratio of 1:1 and molar ratio of 2:1 (ascorbic acid: NaHSeO_3_). The color change from colorless to dark orange indicated the formation of SeNPs.

### 2.3. Implementation of Selenium Nanoparticles (SeNPs) into Leather

Before treatment, the leather samples were scoured with nonionic detergent (3%) on a weight of leather at 50 °C for 15 min at a liquor ratio of 1:50 to remove impurities. Then, the leather sample was rinsed with distilled water, followed by treatment into a solution of SeNPs at a liquor ratio (LR) of 1:50 by ultrasonic water bath. The treatment was carried out at varied temperatures (at room, 40, 50, 60, 65 and 70 °C), different periods (30, 60, 90 and 120 min) as well as different *pH* values (3,4,5,6,7,8) and two concentrations of SeNPs of (25 and 50) mmol/l. Subsequently, the leather sample was removed, rinsed with nonionic detergent (2%) on weight of leather at 40 °C for 15 min at a liquor ratio of 1:50, rinsed with distilled water, and dried in an oven at 50 °C for 25 min. Finally, the leather samples were placed in a desiccator to remove traces of water [4].

### 2.4. Transmission Electron Microscopy (TEM) Analysis

SeNPs characteristics included morphology and size were characterized by transmission electron microscope (JEOL, JEM 2100F) at 200 kV. A drop of the NPs colloidal solution was loaded onto a 400-mesh copper grid with an amorphous carbon film, and water was evaporated in air at room temperature.

### 2.5. Leather Characterization

#### 2.5.1. SEM and EDX Analysis

The surface morphology of the leather samples, including blank samples and those treated with SeNPs, was observed using a scanning electron microscope (JEOL JSM-6510LB, Tokyo, Japan) with attached energy dispersive X-ray spectrum (EDX) unit.

#### 2.5.2. Raman Spectroscopy Analysis

The types of bonds found in the blank or treated leather were detected using a confocal Raman microscope (Jasco NRS-4500, Tokyo, Japan) at the range of 200–1600 cm^−1^. For both Raman data acquisition and processing, Jasco spectroscopy suite software was used.

#### 2.5.3. Colorimetric Study

The color uptake, expressed as the color strength (*K/S*) of blank leather and leather/SeNPs, was determined using a spectrophotometer (CM3600A; Konica Minolta, Japan). *K/S* values were evaluated at the wavelength of maximum absorption (λ_max_) of the color’s reflectance curve at 360 nm. The total color difference (∆*E*) was represented in terms of CIE LAB color space data. It was calculated using (Equation (1)),
(1)$$\Delta E=\sqrt{{\left(L{*}_{2}-L{*}_{1}\right)}^{2}+{\left(a{*}_{2}-a{*}_{1}\right)}^{2}+\left(b{*}_{2}-b{*}_{1}\right){\;}^{2}\;}$$
where, ΔE is the total difference between blank leather and leather/SeNPs, *L** is the lightness from black (0) to white (100), *a** is the red (+)/green (−) ratio and *b** is the yellow (+)/blue (−) ratio.

#### 2.5.4. Exhaustion of SeNPs onto Leather

The UV/Vis spectrum of the SeNPs colloidal solutions before and after leather treatment was measured by UV/Vis spectrophotometer (Alpha-1860, Noble, IN, USA) to evaluate the SeNPs exhaustion.

#### 2.5.5. Physical Properties of Leather

The color fastness of blank leather and leather/SeNPs was tested according to the following standard methods. They were determined using the AATCC (61-1996) [40], (8-1996) [41], and (16-2004) [42] tests for washing, rubbing, and light fastness, respectively. The tensile strength tests of blank leather and leather/SeNPs were performed according to ASTM D638-14 [43], using a tensile testing machine (Zwick Z010, Staufenberg, Germany). Additionally, the durability to washing was evaluated according to AATCC 61(2A)-1996 [44] after five washing cycles.

#### 2.5.6. Cytotoxicity Test of Leather/SeNPs

Cytotoxicity was evaluated on human normal melanocyte cell line (HFB4) using MTT assay [45]. The leather/SeNPs dyed under the optimum conditions were sterilized and cut. Then it was plated on the six-well tissue culture plate. This plate was inoculated with cells and incubated for 24 h at 37 °C. The cell monolayer was washed twice using washing media after growth medium decantation. The cytotoxicity physical signs were checked in the tested cells. The tissue was picked up, and then 20 µL of 3-(4,5-dimethylthiazol-2-yl)-2,5-diphenyltetrazolium bromide dye (MTT) dissolved in phosphate buffer saline at a concentration of 5 mg/mL was added to each well with shaking for 5 min and incubated at 37 °C and 5% CO_2_ for 5 h. The formazan was formed as MTT metabolite and resuspended in 200 µL dimethyl sulfoxide with shaking for 5 min. The absorbance was measured at 560 nm and the background was read at 620 nm, and then subtracted.

#### 2.5.7. Antibacterial Activity

The antibacterial activity test for leather/SeNPs was conducted according to AATCC (147-2004) [46]. The bacterial strains were spread on the media using a sterile cotton swab. Antibacterial activity was evaluated against G+ve bacteria (*Bacillus cereus*) and G-ve bacteria (*Escherichia coli*, *Salmonella typhi and Pseudomonas aeruginosa*). Leather/SeNPs samples were cut to be square in shape and measuring 10 × 10 mm, while standard drugs such as tetracycline and ciprofloxacin were loaded in the disks to compare between their antibacterial activity and that of the tested leather/SeNPs. The petri dishes were then incubated at 37 °C for 24 h. while the growth inhibition zone diameters (mm) were determined.

#### 2.5.8. Statistical Analysis

All tests have been performed by taking the average of three (samples) readings. The standard error of the mean was calculated according to the equation given below and found to be + (−) 0.1
SEX = S/√n
where S = sample standard deviation, and n = the number of observations of the sample

## 3. Results

### 3.1. Transmission Electron Microscopy (TEM) Analysis

The morphology and size of the synthesized SeNPs at different concentrations (25, 50 mM) were characterized using a transmission electron microscope. SeNPs prepared at the concentrations of 25 and 50 mM ranged from 36–77 nm and 41 to 149 nm, indicating the increase in average size with the increase in SeNPs concentration. The obtained TEM micrographs revealed that SeNPs were spherical in shape and well dispersed in the colloidal solution, especially that prepared at the low concentration. Furthermore, no agglomeration or deformation of SeNPs was observed, as displayed in Figure 1a,b. The width of bins in SeNPs histogram was 20 nm, as illustrated in Figure 1c, and these bins were centered at 40, 60, 80, 100, 120 and 140 nm. For example, all particles with diameters between 70 nm and 90 nm were considered together as particles with a diameter of 80 nm. In the case of low concentration (25 mM), SeNPs around 60 nm in size exhibited the highest percentage value of 76.06. On the other hand, the majority of SeNPs at the high concentration of 50 mM were from 70 to 110 nm, which confirmed the increase in SeNPs diameter and polydispersity as the increase in NPs concentration.

Over and above, many SeNPs at the low concentration had a hollow shape (ring-shaped), while SeNPs at the concentration of 50 mM were ordinary solid spherical particles, illustrating that SeNPs specific surface area increased with the decrease in SeNPs concentration due to the reduction in NPs average size and the change in their morphology as shown in Figure 1.

### 3.2. Leather Characterization before and after Functionalization

#### 3.2.1. Scanning Electron Microscopy (SEM) Analysis

The micrographs obtained from SEM analysis showed that the surface of untreated leather is typically clear with clean scales and smooth longitudinal collagen fibers as shown in Figure 2a. On the other hand, the SEM micrographs of leather/SeNPs revealed that the leather surface was coated sufficiently with a layer of SeNPs without aggregation and the SeNPs were distributed well on the leather surface as displayed in Figure 2b,c. Moreover, the chemical elements analysis of leather/SeNPs confirmed the presence of SeNPs on the surface after the dyeing process as shown in Figure 2d.

#### 3.2.2. Raman Analysis

Raman analysis was conducted to confirm the deposition of SeNPs on the dyed leather surface as shown in Figure 3. The comparison between the spectra of leather surface before and after treatment with SeNPs was investigated in the range of 200–1600 cm^−1^. After treatment two intense peaks at 230 and 295 cm^−1^ appeared as a result of SeNPs deposition. Similar results were obtained by Abou Elmaaty et al. [29] who treated the polypropylene fabric with SeNPs and found a peak around 236 cm^−1^ after SeNPs treatment, while Lukács et al. [47] found a pronounced peak at 292 cm^−1^ as an indicator for the presence of SeNPs that confirms the successful deposition of SeNPs on the treated leather surface.

#### 3.2.3. Colorimetric Study

Table 1 showed the (*L**, *a**, *b**, *C**, *h*, *K/S*) values of blank leather and leather/SeNPs. The results revealed that the *K/S* value of leather/SeNPs was significantly higher than the blank leather. The (*L**, *a**, *b**) values for leather/SeNPs were varied considerably from blank leather. The color of blank leather is cream-white with a relatively high *L** value (74.9), low *a** value (8.6), and low *b** value (17.7). The lightness (*L**) value of leather/SeNPs was decreased due to the presence of SeNPs, resulting in the coloration of leather. The *a** values of leather/SeNPs was increased, the increment in *a** value indicated more redness color for the treated leather. The rise in *b** value of leather/SeNPs was observed as yellow/brownish, which confirming the coverage of leather with SeNPs. It can be concluded from the equation 1 and Table 1 that the total color difference (∆*E*) between the two leathers is ~42.

##### Effect of Treatment *pH* on Color Strength (*K/S*)

To investigate the color changes of leather/SeNPs at different *pH* values under an ultrasonic water bath, Figure 4 showed the effect of *pH* on *K/S* of leather/SeNPs. The *pH* was adjusted from 3 to 8 to reflect the different colors expressed in leather/SeNPs. As shown, the maximum *K/S* value appeared at *pH* 6; this might be caused by overcrowding of SeNPs molecules into the leather matrix in this *pH* value due to the ultrasonic power [48]. The results indicated that the *K/S* of leather/SeNPs can be fine-tuned by controlling the *pH*. According to Lee and Kim et al. [49], the correlation between SeNPs and the leather can be attributed to *pH* of treatment bath [48].

##### Effect of Treatment Temperature on Color Strength (*K/S*)

Figure 5 demonstrated the *K/S* values of leathers/SeNPs at different temperatures. The results showed that the maximum *K/S* value was found to be 18.7 at 65 °C. In addition, when the temperature was increased above 65 °C, the *K/S* value was simultaneously decreased. This may be attributed to the tough muscle fiber of leather which did not allow the nanoparticles to enter the leather matrix, causing a lower SeNPs uptake. Moreover, high temperatures can cause the leather to shrink, harden, stiffen, and become brittle. It can be concluded that the temperature of 65 °C was set to be the optimum temperature for binding SeNPs into the leather [48].

##### Effect of Treatment Time on Color Strength (*K/S*)

Figure 6 exhibited the effect of treatment time on SeNPs uptake by leather. As depicted, the *K/S* was increased upon increasing time of treatment. The optimum *K/S* value of 18.7 was achieved at 60 min after which, *K/S* started to decrease with increasing time. The decrease in *K/S* as the increase in time from 90 to 120 min may be due to the aggregation of SeNPs as a result of SeNPs accumulation upon the leather surface [48].

##### Effect of SeNPs Concentration on Color Strength (*K/S*)

The color strength of the leather/SeNPs was found to be dependent on SeNPs concentration. Upon increasing the concentration of SeNPs, the *K/S* value was decreased as shown in Figure 7. The highest *K/S* value for the treated leather was monitored at the lowest concentration. This proved that treating leather with lower concentration of SeNPs (25 mM) was adequate to achieve the optimum *K/S*. The aforementioned behavior could be attributed to the thermal stability of the leather, as it increased at low SeNPs concentration and decreased at high SeNPs concentration. The improvement in thermal stability could be recognized as the crosslinking between SeNPs and collagen of leather [4].

#### 3.2.4. Exhaustion of SeNPs onto Leather

The UV/Visible spectrum of SeNPs colloidal solution was measured before and after leather treatment to evaluate the SeNPs exhaustion by treated leather as shown in Figure 8.

The absorbance of SeNPs was measured in the range of 235 to 700 nm. The absorption spectrum of SeNPs showed a sharp absorption band at 290 nm before the exhaustion, while SeNPs spectrum after leather treatment decreased obviously after exhaustion, confirming the deposition of SeNPs on the treated leather.

#### 3.2.5. Physical Properties of Leather/SeNPs

The tensile modulus and elongation at break % values were listed in Table 2. The obtained results indicated that there were no significant differences about the mechanical properties of leathers, mainly between the leather/SeNPs compared to the blank leather. The color fastness properties of leather/SeNPs were evaluated and the results were tabulated in Table 2. Leather/SeNPs revealed excellent fastness results referring to the chemical stability of the SeNPs onto the leather surfaces along with long-term durable interactions between the SeNPs and the leather surface [4].

#### 3.2.6. Cytotoxicity of Leather/SeNPs

Leather/SeNPs cytotoxicity was evaluated against healthy human melanocyte cell line (HFB4) using MTT assay. The viability of cells in the presence of leather/SeNPs was 90.67% compared to that of negative control. The average relative cell viability is over 70% [50], indicating the low toxicity of leather/SeNPs toward human skin.

#### 3.2.7. Antibacterial Activity

The antibacterial activity of leather/SeNPs was evaluated against four bacterial strains, including *Bacillus cereus* as Gram-positive bacteria in addition to *Escherichia coli*, *Pseudomonas aeruginosa* and *salmonella typhi* as Gram-negative bacteria. The results revealed that the leather/SeNPs showed outstanding antibacterial activity against the tested strains in comparison with standard drugs as listed in Table 3. Leather treated with 25 mM SeNPs colloidal solution was more effective against *Escherichia coli*, *Pseudomonas aeruginosa* and *Bacillus cereus* than that treated with high concentration (50 mM) due to the difference in average size. SeNPs (25 mM) with small average size exhibited higher specific area and in contact with bacterial cells more than that at the high concentration (50 mM) [51]. There was no obvious variation in inhibition zone diameters for leather/SeNPs before and after five washing cycles that confirms the durability of the leather/SeNPs samples at the tested concentrations. Moreover, the cross-linking between leather and SeNPs as antimicrobial agents protects leather against laundering and mechanical abrasion that makes the leather products more durable [22].

The antibacterial effect on bacteria may be due to the release of ions or the formation of reactive oxygen species that lead to DNA damage. Furthermore, the deposited SeNPs can contact bacteria or fungi cells much easier than colloidal form [22].

## 4. Conclusions

Leather material was successfully colored and functionalized utilizing selenium nanoparticles (SeNPs) which were synthesized and deposited simultaneously onto the leather surface. The SeNPs decorated the leather surface with shining colors, which can be controlled by adjusting the *pH* at 6, treatment time for 60 min., treatment temperature at 65 °C and SeNPs concentration of 25 mM. The results showed that a yellow/brown color was imparted to the leather after the implementation with SeNPs by ultrasound technique. Moreover, the colored leather samples acquired good color fastness to rub, wash, and light. Leathers/SeNPs exhibited excellent antibacterial activities against models of bacteria, including Gram-positive bacteria (*Bacillus cereus*) and Gram-negative bacteria (*Pseudomonas aeruginosa*, *Escherichia coli* and *Salmonella typhi*). The results of coloration, cytotoxicity and antibacterial properties clarified that the SeNPs can be used to impart color and antibacterial properties to leather material. The proposed methodology emphasized the effectiveness and applicability of this simple approach to the footwear industry to color the leather as well as prevent the spread of bacterial infection promoted by humidity, poor breathability and temperature.

## Figures and Tables

**Figure 1 polymers-14-00074-f001:**
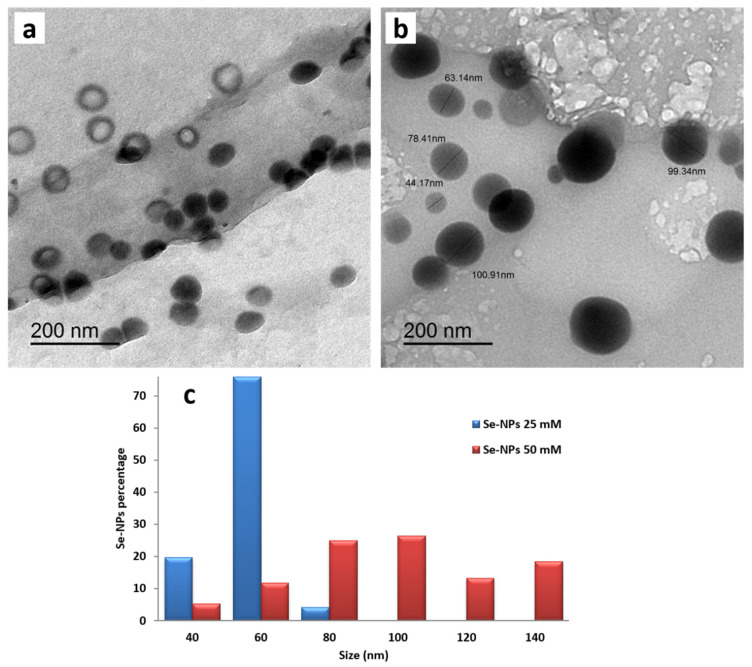
TEM images of SeNPs synthesized at different concentrations of (**a**) 25 mM and (**b**) 50 mM in addition to (**c**) the size distribution histogram of the prepared SeNPs.

**Figure 2 polymers-14-00074-f002:**
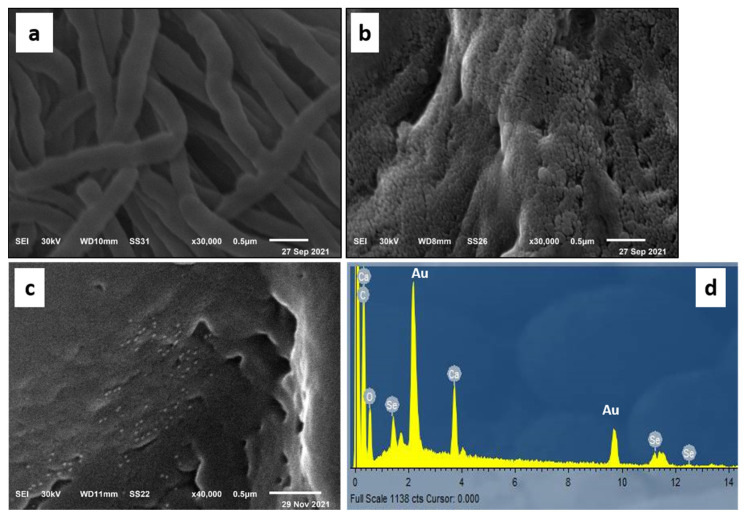
SEM micrographs of (**a**) blank leather and (**b**,**c**) leather dyed with SeNPs under the optimum conditions at different scales as well as (**d**) EDX spectrum of the dyed leather.

**Figure 3 polymers-14-00074-f003:**
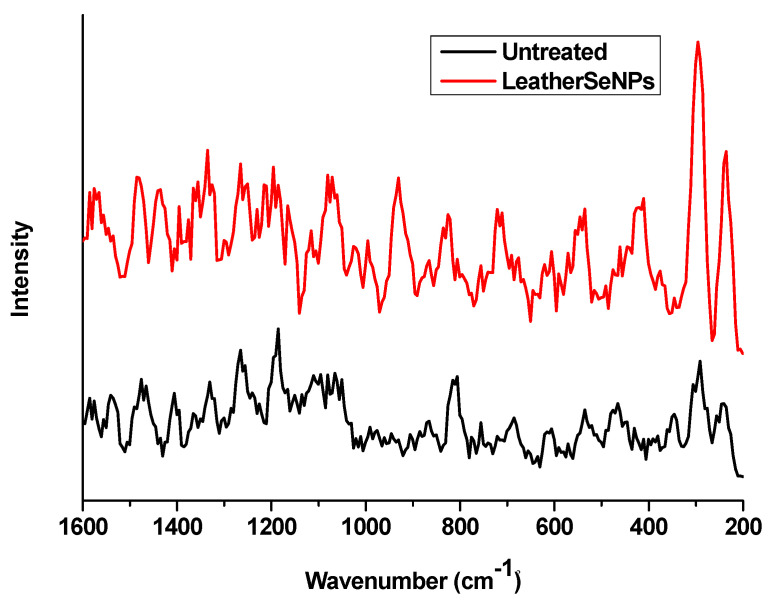
Raman spectra of untreated leather and leather/SeNPs⁻.

**Figure 4 polymers-14-00074-f004:**
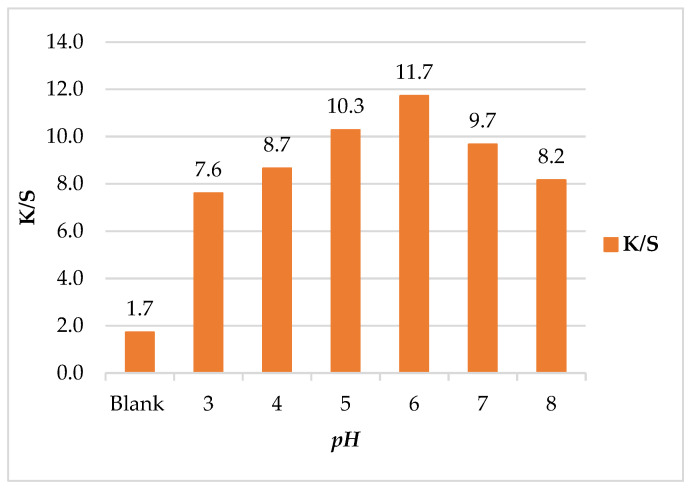
Effect of *pH* on color strength (*K/S*) of leather/SeNPs.

**Figure 5 polymers-14-00074-f005:**
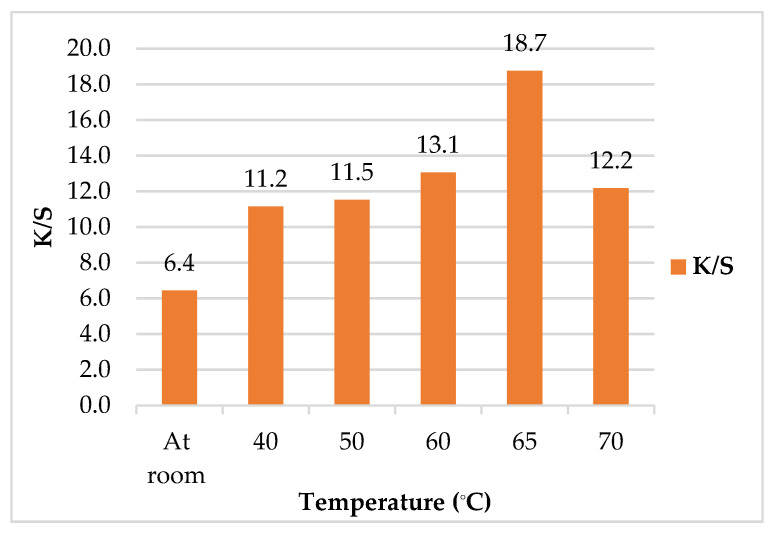
Effect of treatment temperature on color strength (*K/S*) of leather/SeNPs.

**Figure 6 polymers-14-00074-f006:**
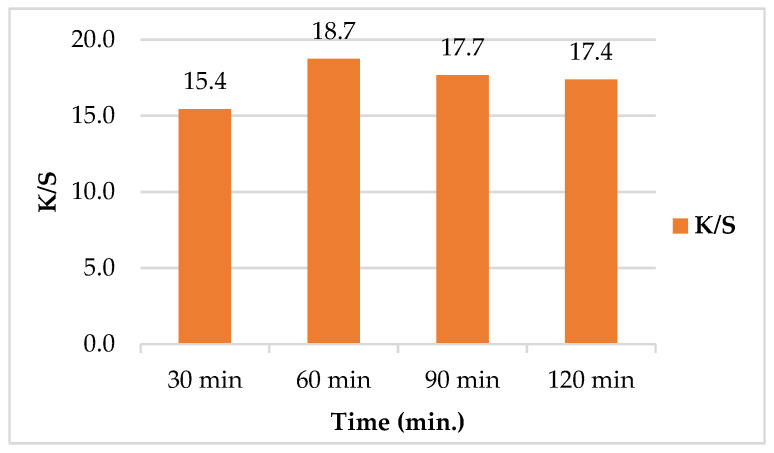
Effect of treatment time on color strength (*K/S*) of leather/SeNPs.

**Figure 7 polymers-14-00074-f007:**
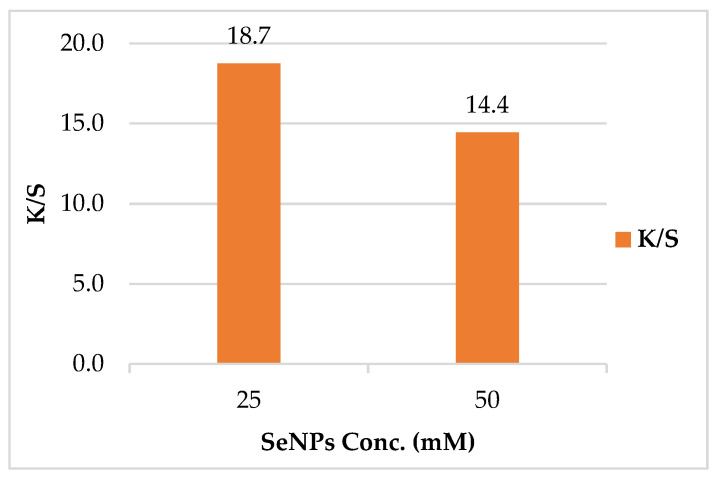
Effect of SeNPs concentrations on color strength (*K/S*) of leather/SeNPs.

**Figure 8 polymers-14-00074-f008:**
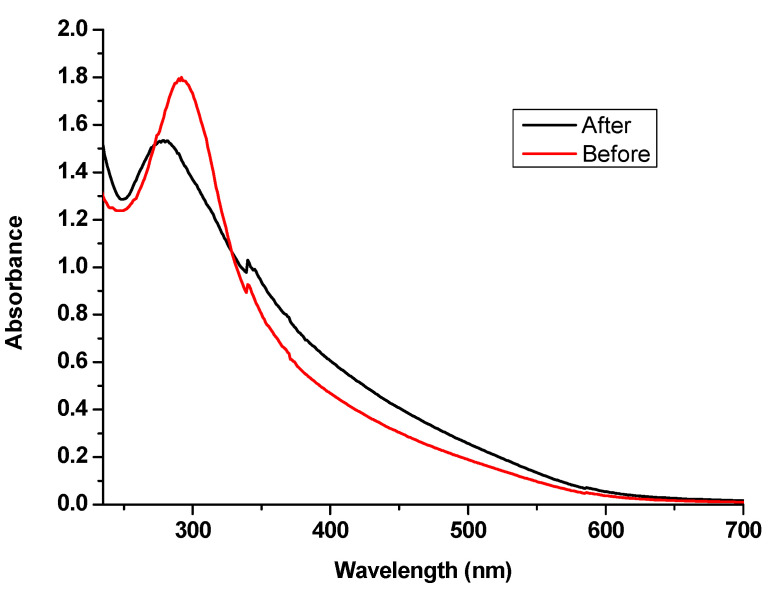
UV/Visible spectrum of SeNPs before and after leather treatment.

**Table 1 polymers-14-00074-t001:** Color strength and colorimetric data of blank leather and leather/SeNPs.

Type	Sample	Color Parameters
*L**	*a**	*b**	*C**	*h*	*K/S*
Blank leather	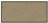	74.9	8.6	17.7	22.5	64.0	1.7
Leather/SeNPs	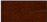	37.1	26.1	24.8	19.6	43.5	18.7

**Table 2 polymers-14-00074-t002:** Properties of the Leather and Leather/SeNPs under optimum conditions.

Sample	Wash Fastness		Rubbing Fastness		Light Fastness	Tensile Strength	
	St.	Alt.	Dry	Wet		Tensile Modulus, ^a^ MPa	Elongation, ^b^ %
Blank leather	-	-	-	-	-	8	36
Leather covered with SeNPs	5	5	5	4/5	4/5	7	32
Leather covered with SeNPs after 5 washing cycles (durability test)	5	5	5	4/5	4/5	7	32

Treatment conditions: SeNPs, (25 mM); Time (60) min; Temp. (65) °C; *pH* 6. ^a^ Standard deviation for untreated leather = 0.5, standard deviation for treated leather = 0.5. ^b^ Standard deviation for untreated leather = 0.5, standard deviation for treated leather = 0.5.

**Table 3 polymers-14-00074-t003:** The inhibition zone (mm) values of leathers/SeNPs treated with different SeNPs concentrations.

Substrate	*Bacillus* *cereus* (G+)	*Escherichia coli* (G−)	*Pseudomonas* *aeruginosa* (G−)	*Salmonella* *typhi* (G−)
Leather/SeNPs (25 mM)	20	15	19	16
Leather/SeNPs (50 mM)	18	9	15	18
Leather/SeNPs (25 mM); after 5 washing cycles	20	14	18	14
Leather/SeNPs (50 mM); after 5 washing cycles	17	8	15	18
Tetracycline (30 µg)	15	19	16	13
Ciprofloxacin (10 µg)	18	21	17	15

## Data Availability

The data presented in this study are available on request from the corresponding author.

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
