# Peer review of "Simultaneous Sonochemical Coloration and Antibacterial Functionalization of Leather with Selenium Nanoparticles (SeNPs)"

_polymers, 2021, doi:10.3390/polym14010074_

Round 1

Reviewer 1 Report

 This work focuses on the fabrication of a leather material taking advantage of selenium nanoparticles (SeNPs) intrinsic antibacterial activity and its coloring ability.

This is an interesting manuscript ; Nevertheless some minor revisions are needed in order to publish this work.

  1. Could the authors provide some valid references in sections 2.2 and 2.3?
  2. Could the authors comment about the re-usability of their samples regarding their antibacterial activity?
  3. Section 2.5.6 needs further analysis and details. 
  4. A few typos and syntax errors should be corrected.

Author Response

Comment (1)

Could the authors provide some valid references in sections 2.2 and 2.3?

Reply

references have been added to sections 2.2 and 2.3.

Comment (2)

Could the authors comment about the re-usability of their samples regarding their antibacterial activity?

Reply

a durability test has been for the treated samples and the results are presented in Table 3.

Comment (3)

Section 2.5.6 needs further analysis and details. 

Reply

The method was written in details as required with the dimensions of the tested leather.

Comment (4)

A few typos and syntax errors should be corrected.

Reply

the manuscript has been edited to correct these errors as required.

Reviewer 2 Report

The paper entitled “Simultaneous sonochemical coloration and antibacterial functionalization of leather with selenium nanoparticles (SeNPs)” focuses on an ultrasonic technique used for simultaneous coloration and functionalization of leather via implementation of SeNPs. The obtained SeNPs coated leather have been examined by TEM, SEM with EDX, colorimetric studies, UV/Vis spectroscopy, tensile strength test, cytotoxicity test, and antibacterial activity. The tests used are versatile and complementary while the obtained results are interesting and promising. In my opinion, the paper should be interesting from a scientific and practical point of view. However, I would like to recommend the publication of the manuscript in this journal after fulfilling the following recommendations:   

  1. In fig. 1, micrographs a and b present one and the same image at slightly different magnification, so one of them can be omitted. In contrast, higher magnification of the SeNPs can be used to illustrate the coating morphology. Moreover, all peaks in the EDX spectrum should be identified.
  2. Section 3.2.4. is improperly entitled. Moreover, it is not clear for the readers which samples have been tested. The effect of low-pH (pH 3 and 4) and high-temperature treatment on the mechanical properties and durability of the SeNPs coated leather should also be considered and the results should be presented in this study.
  3. Most of the results presented (e.g. those in Tables 2 and 3) seem statistically unreliable. It is unclear how many samples have been tested.
  4. It is recommendable the authors include Fourier transform infrared spectroscopy (FTIR) analysis in their study to characterize the functional groups that participate in the bonding of SeNPs with leather;
  5. The durability of the examined SeNPs coating on the leather surface should be further discussed. The results can be compared with similar studies.
  6. The antibacterial activity of the SeNPs coated leather should be compared with a control sample for accurate comparison. Otherwise, the results seem unreliable.

Author Response

Comment (1)

In fig. 1, micrographs a and b present one and the same image at slightly different magnification, so one of them can be omitted. In contrast, higher magnification of the SeNPs can be used to illustrate the coating morphology. Moreover, all peaks in the EDX spectrum should be identified.

Reply

It seems that this comment is related to Figure 2 not Figure 1 because in Fig .1 there are two different micrographs for varied Se-NPs concentration 25 mM (a) and 50 mM (b). In addition, EDX spectrum was shown in Fig. 2.

Regarding Fig. 2, micrograph (b) was omitted as required. In addition, higher magnification was used to illustrate the size and morphology of SeNPs deposited on leather surface as shown in micrograph (c).

The peaks around 2.3 and 9.8 KeV were corresponding to Au used in the sample preparation for SEM and EDX analysis for high resolution imaging.

Comment (2)

Section 3.2.4. is improperly entitled. Moreover, it is not clear for the readers which samples have been tested. The effect of low-pH (pH 3 and 4) and high-temperature treatment on the mechanical properties and durability of the SeNPs coated leather should also be considered and the results should be presented in this study.

Reply

As for the sample on which the tests were conducted: the treatment conditions have been added to Table 2 caption.

the reason of not mentioning the effect of low-pH (pH 3 and 4) and high-temperature treatment on the mechanical properties; is due to the fact that at low pH such as 3 and 4, the selenium nanoparticles uptake by leather was so small which has been proven by the K/S values as well;

the use of high temperatures caused severe hardening and consequently significant deterioration of the sample.

As for the durability of physical properties; the results have been added in Table 2 after 5 washing cycles (durability test).

Comment (3)

Most of the results presented (e.g. those in Tables 2 and 3) seem statistically unreliable. It is unclear how many samples have been tested.

Reply

statistical analysis has been added where the test was conducted on 3 samples.

Comment (4)

It is recommendable the authors include Fourier transform infrared spectroscopy (FTIR) analysis in their study to characterize the functional groups that participate in the bonding of SeNPs with leather;

Reply

In this respect, we used Raman analysis instead of I.R to distinguish the bond between selenium metal and leather surface because the selenium peak can be observed in Raman analysis easier than I.R spectra.

Comment (5)

The durability of the examined SeNPs coating on the leather surface should be further discussed. The results can be compared with similar studies.

Reply

The durability was discussed in details especially in section 3.3.7. The leather/SeNPs samples were tested before and after washing (5 cycles) to confirm the durable antibacterial activity of the tested leather/SeNPs. Also, we mentioned a study related to the antibacterial activity of leather and its role in the durability.

Comment (6)

The antibacterial activity of the SeNPs coated leather should be compared with a control sample for accurate comparison. Otherwise, the results seem unreliable.

Reply

Thank you for this valuable advice, standard drugs such as tetracycline and ciprofloxacin were used as a control to compare the obtained results with them as listed in Table 3

Round 2

Reviewer 2 Report

The authors have carefully addressed all the review’s recommendations and the manuscript has been substantially improved.